# Novel Salinity-Tolerant Third-Generation Hybrid Rice Developed via CRISPR/Cas9-Mediated Gene Editing

**DOI:** 10.3390/ijms24098025

**Published:** 2023-04-28

**Authors:** Xiabing Sheng, Zhiyong Ai, Yanning Tan, Yuanyi Hu, Xiayu Guo, Xiaolin Liu, Zhizhong Sun, Dong Yu, Jin Chen, Ning Tang, Meijuan Duan, Dingyang Yuan

**Affiliations:** 1National Center of Technology Innovation for Saline-Alkali Tolerant Rice, Hunan Hybrid Rice Research Center, Changsha 410125, China; 2College of Bioscience and Biotechnology, Hunan Agricultural University, Changsha 410128, China; 3Hunan Academy of Agricultural Sciences, Changsha 410125, China; 4Sanya National Center of Technology Innovation for Saline-Alkali Tolerant Rice, Sanya 572019, China

**Keywords:** CRISPR/Cas9, heterosis utilization, *OsRR22*, rice, salinity tolerance

## Abstract

Climate change has caused high salinity in many fields, particularly in the mud flats in coastal regions. The resulting salinity has become one of the most significant abiotic stresses affecting the world’s rice crop productivity. Developing elite cultivars with novel salinity-tolerance traits is regarded as the most cost-effective and environmentally friendly approach for utilizing saline-alkali land. To develop a highly efficient green strategy and create novel rice germplasms for salt-tolerant rice breeding, this study aimed to improve rice salinity tolerance by combining targeted CRISPR/Cas9-mediated editing of the *OsRR22* gene with heterosis utilization. The novel alleles of the genic male-sterility (GMS) and elite restorer line (733S*^rr22^*-T1447-1 and HZ*^rr22^*-T1349-3) produced 110 and 1 bp deletions at the third exon of *OsRR22* and conferred a high level of salinity tolerance. Homozygous transgene-free progeny were identified via segregation in the T2 generation, with *osrr22* showing similar agronomic performance to wild-type (733S and HZ). Furthermore, these two *osrr22* lines were used to develop a new promising third-generation hybrid rice line with novel salinity tolerance. Overall, the results demonstrate that combining CRISPR/Cas9 targeted gene editing with the “third-generation hybrid rice system” approach allows for the efficient development of novel hybrid rice varieties that exhibit a high level of salinity tolerance, thereby ensuring improved cultivar stability and enhanced rice productivity.

## 1. Introduction

With the rapidly increasing population, changing diet, and increasing biofuel consumption, global crop productivity must double by 2050 [1]. However, abiotic stressors, including soil salinity, drought, cold, and heat can severely limit the feasibility of yield increase or even reduce crop productivity in large areas [2]. Salinity is a major limiting factor in crop productivity as most crops are not salt-tolerant [3]. The land affected by salt stress is also increasing for various reasons, including tsunamis, sea-level rise, and climate change [4]. For example, the 2011 earthquake and tsunami in Japan led to the flooding of more than 20,000 ha of rice paddy fields with seawater, resulting in increased soil salinity [5]. Over 800 Mha of arable land (approximately 6% of the world’s total land used for cropping) is affected by salinity to varying degrees [3]. Therefore, developing salt-tolerant crop varieties is a promising strategy to mitigate the negative impact of salinity on crop yields.

Rice (*Oryza sativa* L.) is the dominant food crop in Asia and worldwide. However, rice is sensitive to salt stress [6,7], displaying reduced grain yields under salt stress. Many quantitative trait loci (QTLs) and genes associated with regulating salt stress tolerance in rice have already been mapped or cloned, including the NHX family (*OsNHX1*, *OsNHX2, OsNHX3*, and others) [8,9,10], HKT family (Os*HKT1*, Os*HKT2*, *OsHKT7*, and others) [11,12,13], *OsKAT1* [14], *OsbZIP71* [15], *OsSALP1* [16], *OsBADH1* [17], *OsNAC5* [18], *DST1* [19], *DCA1* [20], *OsRR22* [5], and many others [21]. However, few of these genes have been utilized to develop elite or hybrid rice varieties in particular. The rice *RR22* gene (*OsRR22*) is a significant gene located on chromosome 6 that is a negative regulator of salinity tolerance and encodes a 696-amino acid B-type response regulator transcription factor, which is involved in both cytokinin signal transduction and metabolism. As previously reported, the loss of its function results in rice with a significantly improved salinity tolerance [5].

Genome-editing technologies provide an efficient method for creating nucleotide mutations in target genes. Zinc finger nucleases (ZFNs), transcription activator-like effector nucleases (TALENs), and clustered regularly interspaced short palindromic repeats (CRISPR)/CRISPR-associated (Cas) 9 (CRISPR/Cas9) have been previously widely used in precision crop breeding [22,23,24,25,26,27,28,29], including for salinity tolerance. For instance, CRISPR/Cas9-mediated mutagenesis of the *OsRR22* gene has been used to create salinity-tolerant Japonica rice mutants [30]. However, this method has only been performed in the japonica cultivar WPB106 background, and it was not utilized with indica rice varieties, particularly hybrid rice elite parental varieties.

Hybrid vigor, also known as heterosis, is a widely observed phenomenon in crops, with the development and adoption of hybrids contributing significantly to global agricultural productivity. Hybrid rice, for instance, can yield 10–20% more than conventional rice [31,32]. Indica hybrids are currently the primary crops utilized for commercial rice production, especially in Southeast Asia and Southern China, where coastal lowlands are threatened by soil salinity due to seawater intrusion [33]. Hence, the development of genetically modified indica rice with salinity tolerance has become a critical priority in recent years. Prof. Yuan Longping demonstrated in 2017 that combining heterosis utilization with salt-tolerant genes can be utilized to develop salt-tolerant rice varieties. The “third-generation hybrid rice breeding system” has recently been used to create elite hybrid rice. This hybrid rice breeding technology combines the benefits of the stable sterility of the cytoplasmic male-sterility system and the free combination of the photoperiod/thermo-sensitive genic male-sterility system [34]. 

Therefore, in this study, CRISPR/Cas9-mediated editing and third-generation hybrid rice system approaches were utilized to create *OsRR22* targeted mutations in the genic male-sterility (GMS) line (733S*^rr22^*-T1447-1) and restorer line (HZ*^rr22^*-T1349-3). These *rr22* plants exhibited significantly enhanced salinity tolerance without any apparent changes to other major agronomic traits relative to wild-type (733S, HZ) under normal conditions. Further, the novel salt-tolerant promising third-generation hybrid rice line ST1 was created by crossing with the *rr22* mutant parents. The aim of our study was to offer a new, green, highly efficient strategy and novel rice germplasms for salt-tolerant rice breeding. It may become one of the most significant ways/materials to ensure food security by utilizing the vast saline-alkali land.

## 2. Results

### 2.1. Development of OsRR22 Mutagenized Plants

CRISPR/Cas9-based genome editing was employed to generate *OsRR22* mutagenesis lines. The third exon of OsRR22 was targeted using three gRNAs designed for the 22KT1, 22KT2, and 22T3 target sites (depicted in Figure 1A). These sites were chosen because they are located near an SNP in the third exon of *OsRR22*, which was reported to be associated with increased tolerance to salinity [5]. Two different CRISPR/Cas9 vectors, including one containing two gRNAs (pC-22-KT12) and another containing a single gRNA (pC-22-T3), were constructed (Figure 1B) [35]. These vectors were inserted into 733B and HZ varieties via *Agrobacterium*-mediated transformation. All T0 plants were tested for the presence of T-DNA using *HPT*-specific PCR primers HPT-F1/R1 (Appendix A). A target-specific PCR primer pair RR-CJ3F/RR-CJ3R (Appendix A) surrounding the target region was used to amplify the transgenic plants, and the resulting PCR products were Sanger-sequenced to determine the *OsRR22* mutagenesis of the T0 generation plants. Sequencing revealed 21 and 10 edited plants in the 733B and HZ backgrounds, respectively, with an editing efficiency of 84% (21/25) and 58.82% (10/17). These results demonstrated that these pC-22-KT12 and pC-22-T3 constructs could rapidly generate *OsRR22*-mutagenized rice plants.

### 2.2. Characterization of OsRR22 T0 Mutant Plants

Ten (pC-22-T3, HZ) T0 *osrr22* plants were selected to analyze the target site mutations. These genotypes were divided into two categories (Figure 2A): (1) Homozygotes or bi-allelic (editing efficiency: 90%, 9/10, e.g., T1347 or T1349 at T3); (2) chimeras (editing efficiency: 10%, 1/10, e.g., T1352 at T3). Furthermore, only one type of mutation was identified: Deletions. Single-nucleotide deletions were the most common forms of mutations, accounting for 55.56% of all mutations (Figure 2B).

### 2.3. Development of Transgene-Free Mutant and Homozygous Osrr22 Lines

To develop homozygous *osrr22* lines free of T-DNA, ten edited lines from T0 progeny (five lines from 733B and five lines from the HZ background) were bred to the T1 generation. All T1 plants from the selected, edited lines were further tested for T-DNA and mutation using *HPT*- and *Cas9*-specific PCR primers and RR-CJ3F/RR-CJ3R target-specific PCR primers (Appendix A). Transgene-free plants were those in which neither *Cas9* nor *HPT* was detected. Such plants were detected at frequencies of 0.00 to 33.33% among the analyzed plants (Appendix A and Figure 3). Furthermore, three (733B background) and four (HZ background) transgene-free plants bearing homozygous *osrr22* were selected. Seeds of these verified, transgene-free, homozygous mutant plants (designated as 733B*^rr22^*-T1430-7, 733B*^rr22^*-T1431-5, 733B*^rr22^*-T1447-1, HZ*^rr22^*-T1347-6, HZ*^rr22^*-T1349-3, HZ*^rr22^*-T1350-2, and HZ*^rr22^*-T1351-8) were harvested and used to generate the T2 progeny. In addition, the genetic segregation analysis of self-pollination of three mutagenesis maintainer lines (733B*^rr22^*-T1430-7, 733B*^rr22^*-T1431-5, and 733B*^rr22^*-T1447-1) demonstrated fluorescent and nonfluorescent seeds (Figure 4). The nonfluorescent seeds were selected to generate a T2 population as the GMS lines (named 733S*^rr22^*-T1430-7-T2, 733S*^rr22^*-T1431-5-T2, and 733S*^rr22^*-T1447-1-T2), transgene-free and homozygous for *osrr22*. 

While powerful, CRISPR/Cas9 technology has been reported to be susceptible to introducing off-target mutations [36,37]. We, therefore, assessed the off-target efficiency of 22KT1, 22KT2, and 22T3 at high-sequence-similarity sites (<5 mismatch bp), with PAM, and in the coding sequence (CDS) or intron, to our target sites. Ten different potential off-target sites within the rice genome were examined. However, no evidence of off-target events among the 180 T1 plants was detected (Appendix A).

### 2.4. Novel OsRR22 Alleles Conferring Salt Tolerance

All transgene-free and homozygous *osrr22* T3 lines and wild-type (733S and HZ) seedlings were treated with a fresh Yoshida nutrient solution or a 0.8% NaCl nutrient solution at the three-week-old seedling stage. After treatment with the fresh Yoshida nutrient solution for 10 days, there was no significant difference observed between *osrr22* and wild-type plants (Figure 5A). However, phenotypic damage, including leaf rolling, whitish and brownish leaf tips, drying leaves, and a reduction in leaf/root growth and fresh height, was observed in wild-type plants, with similar salt damage detected in *osrr22* after 10 days of 0.8% NaCl treatment (Figure 5B). Notably, *osrr22* plants showed better growth than wild-type plants. After receiving the Yoshida nutrient solution for 7 days, most *osrr22* seedlings recovered and displayed continuous growth, while the wild-type seedlings remained severely damaged (Figure 5C,D). After subjecting the rice seedlings to 10 days of salt treatment and 7 days of recovery, it was observed that three *osrr22* lines (733S*^rr22^*-T1447-1-T3, -110bp, HZ*^rr22^*-T1347-6-T3, -A, HZ*^rr22^*-T1349-3-T3, -G) exhibited significantly increased fresh weight, seedling length, and root length values relative to wild-type (Table 1). 733S*^rr22^*-T1447-1-T3 and HZ*^rr22^*-T1349-3-T3 displayed the best growth among the mutagenesis lines (Figure 5 and Table 1). The fresh weight of 733S*^rr22^*-T1447-1-T3 and HZ*^rr22^*-T1349-3-T3 increased by 2.68 and 3.87 times [(0.92 − 0.25)/0.25, (0.73 − 0.15)/0.15], respectively, compared to the wild-type (Table 1). Similarly, the seedling length of 733S*^rr22^*-T1447-1-T3 and HZ*^rr22^*-T1349-3-T3 increased by 32.99% and 34.66% [(22.17 − 16.67)/16.67, (29.53 − 21.93)/21.93], respectively (Table 1). Following the 0.8% NaCl treatment for 10 days, the dry weight of the wild-type was lower by 166.49% and 224.56% [(183.00 − 68.67)/68.67, (185.00 − 57.00)/57.00], respectively, compared with 733S*^rr22^*-T1447-1-T3 and HZ*^rr22^*-T1349-3-T3 (Table 1). Meanwhile, the survival ratios of 733S*^rr22^*-T1447-1-T3 and HZ*^rr22^*-T1349-3-T3 significantly increased, compared to the wild-type after recovery with the fresh Yoshida nutrient solution for 7 days. The difference analysis of the fresh weight, seedling length, root length, dry weight, and survival ratios indicated that 733S*^rr22^*-T1447-1-T3 and HZ*^rr22^*-T1349-3-T3 were significantly different from wild-type plants. These results demonstrate the development of novel salt-tolerant mutants in rice via CRISPR/Cas9-mediated editing of the *OsRR22* gene.

### 2.5. Analysis of the OsRR22 Amino Acid Sequence of 733S^rr22^-T1447-1 and HZ^rr22^-T1349-3 Alleles

The OsRR22 amino acid sequences from the 733S*^rr22^*-T1447-1, HZ*^rr22^*-T1349-3, and wild-type (733S and HZ) plants were aligned using DNAMAN5.2.2. The results demonstrated that the two mutagenesis genotypes encoded OsRR22, 187 or 116 amino acids in length, respectively. In contrast, the wild-type (733S and HZ) OsRR22 was 695 amino acids in length (Figure 6). The amino acid sequences in these novel OsRR22 alleles were altered, emphasizing that these mutant alleles expressed truncated, disrupted, and altered OsRR22 proteins.

### 2.6. Agronomic Trait Analysis of 733S^rr22^-T1447-1, HZ^rr22^-T1349-3

The T4 population carrying the homozygous 733S*^rr22^*-T1447-1 and HZ*^rr22^*-T1349-3 alleles and wild-type (733S and HZ) were cultivated under natural growth conditions to evaluate the effects of the 733S*^rr22^*-T1447-1 and HZ*^rr22^*-T1349-3 alleles on the major agronomic traits. No significant differences were detected in the plant height, the number of tillers per plant, the number of grains per panicle, spikelet fertility, or 1000-seed weight at the maturation stage between *osrr22* and wild-type (Table 2 and Figure 7). Therefore, these findings suggest that the 733S*^rr22^*-T1447-1 and HZ*^rr22^*-T1349-3 alleles did not affect the major agronomic traits when the plants were cultivated under natural growth conditions.

### 2.7. Development of the New Promising Third-Generation Hybrid Rice Line with Novel Salt Tolerance

In order to develop the novel salt-tolerant third-generation hybrid rice line, the 733S*^rr22^*-T1447-1-T4 (as the female parent) and HZ*^rr22^*-T1349-3-T4 (as the male parent) were crossed, and the promising hybrid line was named ST1. The salt-tolerant hybrid rice cultivar Xiang Liang You 900 (XLY900) was used as a control (CK). Whole growth stage trials were conducted in a salt-treated pool during the 2022 growing season to assess the practical applicability of ST1 for salt-tolerant rice breeding. Following the local farmers’ cultivation practice, four-week-old ST1 and CK seedlings were transplanted and cultivated. Plants in the high-salinity plot were treated by periodically supplying water containing seawater (an ultimate density of approximately 0.6% NaCl concentration) until the maturation stage. The soil salinity level in the salt-treated plot was monitored by measuring the EC values in the soil. Before the end of February, the EC value was <1 dS/m, but the EC value gradually increased over time. In late March, the EC value exceeded 6 dS/m (equivalent to approximately 0.5% salinity) (Figure 8C). During the whole growth stage trial, the grain yield per ST1 plant in salt-treated plots was significantly increased to 16.22 ± 0.61 g (equivalent to 3.65 tons/hectare, calculated at approximately 225,000 plants per hectare), demonstrating a production increase of more than 45% when compared to the CK plants (Figure 8B). All ST1 plants in the salt-treated plots survived until the end of the trial (Figure 8A). These findings provide further evidence that the CRISPR/Cas9-mediated mutagenesis of the *OsRR22* gene combined with the “third-generation hybrid rice system” can be used to produce promising third-generation hybrid rice varieties with novel salt tolerance. This strategy could increase grain yield by enabling the cultivation of salt-tolerant varieties in salt-contaminated land.

## 3. Discussion 

Salinity is a major abiotic stress that negatively affects crop growth and production. Crop productivity in rice is severely limited due to the high salt concentrations in the soil. Generally, genetic improvement is deemed a cost-effective and environmentally friendly means to enhance salinity tolerance in rice. In recent years, many genes involved in salinity tolerance have been identified in rice, including *OsNHX1, SKC1*, *OsbZIP71*, and *OsNAC2* [8,13,15,38]. Nonetheless, most of these positively regulate the salinity tolerance of rice, and over-expressing these genes shows improved tolerance to salt stresses. However, transgenic crops generated by gene addition are subjected to rigorous genetically modified management. The latest breeding strategy to knock down or knock out a negatively regulated gene is an alternative approach for the genetic improvement of rice and avoiding the transgenic issue using CRISPR/Cas9 technology. Nowadays only a few genes acting as negative regulators of salt tolerance in rice have been identified (e.g., the *OsRR22* gene). A previous study suggested that the *OsRR22* loss-of-function mutation showed significantly enhancing salinity tolerance but does not affect other major agronomic traits [5]. Therefore, this gene has great potential for genetic improvements via CRISPR/Cas9-mediated editing.

In this study, the pC-22-KT12 and pC-22-T3 Cas9 vectors were used to edit *OsRR22*, achieving 84.00% and 58.82% editing efficiency, respectively, in T0 transgenic plants. By transgene self-pollination segregation, seven independent *osrr22*-mutagenesis lines were generated bearing T-DNA-free and homozygous *OsRR22* mutations in the T2 progeny. The transgene-free and *rr22* GMS lines were obtained via fluorescent sorting. The salinity tolerance evaluations at the seedling stage demonstrated that the salinity tolerance was markedly enhanced in two novel *OsRR22* alleles (733S*^rr22^*-T1447-1-T3 and HZ*^rr22^*-T1349-3-T3), suggesting that these alleles are valuable resources for developing elite rice varieties that can be cultivated on salt-contaminated soil. Further analysis of the *OsRR22* translation, OsRR22 amino acid sequences of 733S*^rr22^*-T1447-1, HZ*^rr22^*-T1349-3, and wild-type lines demonstrated that OsRR22 proteins were truncated, altered, and disrupted in these particular alleles. The previous study indicated that an SNP representing a nonsense mutation of TGG to TAG resulting in a loss-of-function mutation in *OsRR22* is responsible for the salinity-tolerance phenotype [5]. Therefore, the 110 bp and 1 bp deletions at 22KT12 and 22T3 caused a frameshift and resulted in premature translation termination and may lead to the enhanced salinity tolerance of 733S*^rr22^*-T1447-1 and HZ*^rr22^*-T1349-3, respectively. In summary, a loss of function of the OsRR22 protein was the cause of the destruction of the two-component cytokinin signaling in these *rr22* plants, resulting in the enhanced salinity-tolerance phenotype. Furthermore, a new promising third-generation hybrid rice line (ST1) with a novel salt-tolerant phenotype was developed by crossing the 733S*^rr22^*-T1447-1-T4 and the HZ*^rr22^*-T1349-3-T4 lines. The ST1 line exhibited favorable agronomic traits in salt stress, was transgene-free, and could be readily used for farming production if permitted under government policies. These results highlight the potential to use CRISPR/Cas9 technology to develop promising hybrid rice lines and parental varieties bearing targeted mutations capable of improving the salinity-tolerance trait, with *OsRR22* being a putatively ideal target for the accelerated development of novel salt-tolerant rice varieties.

Technological advances in the agricultural sector, particularly breeding techniques, have increased agricultural productivity [39,40,41,42]. CRISPR/Cas9 technology has recently become a useful tool for editing genes owing to its simple methodology, mature technology, high mutation efficiency, and ability to avoid the transgenic issue [43]. It has been used in many plant species to improve major agronomic traits, including yield traits, fertility, disease resistance, and others [44,45,46,47,48]. Meanwhile, the development and adoption of hybrids for several key crops have significantly contributed to increased global agricultural productivity since the 1960s. As the latest rice heterosis utilization technology, the “third-generation hybrid rice system” holds great promise to boost hybrid rice production further. The previous report suggests that a promising third-generation hybrid rice combination exhibits more than a 13% yield advantage over the control combinations [49]. Therefore, the CRISPR/Cas9 approach was combined with a “third-generation hybrid rice system” to rapidly produce one new promising hybrid rice line with a novel salt-tolerant phenotype with desirable agronomic traits. The positive results suggest that integrating the CRISPR/Cas9 approach and heterosis utilization may represent a powerful, highly efficient, and green approach to rice and other hybrid crop breeding genetic improvement. Since environmental deterioration is extensive throughout salt-contaminated paddy fields, the resulting crop productivity in rice is seriously limited. Therefore, enhanced rice salinity tolerance is one of the best strategies to expand the rice planting area, ensuring better stability and improved rice yield. As a result, this study offers a new strategy and novel *OsRR22* alleles for breeding new rice varieties with novel salt tolerance and favorable agronomic traits under the conditions of severe salt pollution. 

## 4. Materials and Methods

### 4.1. Plant Materials and Wild-Type Growth Conditions

*Oryza sativa* L. Indica lines 733-3B and HZ that had stable inheritance and were easily distinguishable were used in this study. 733-3B is a maintainer of the GMS 733-3S. HZ is an elite restorer line of hybrid rice. Plants were cultured at 28–35 °C in the greenhouse in Changsha or were grown in the screen house of the Hunan Hybrid Rice Research Centre in Changsha or Sanya during the typical rice-growing season. Management followed standard agricultural practices.

### 4.2. Vector Construction and Transformation

The CRISPR/Cas9 system was used to generate knockout mutants. For high-efficiency and targeted mutagenesis of 733-3B, two candidate sites for *OsRR22* (Os06g0183100) were selected as target single guide RNAs (gRNAs): 22KT1 (5′-tgatcagaagaaccacaggt*AGG*-3′) and 22KT2 (5′-ctgggcttctttgcagctga*GGG*-3′) with the anti-sense strand sequence containing the PAM motif 20 bp downstream. Another target gRNA 22T3 (5′-agacaaagactgtgatgaag*GGG*-3′) was selected for targeted mutagenesis in HZ. The specificity of the target gRNAs was evaluated using biology-website CRISPR-GE (http://skl.scau.edu.cn/) and NCBI (https://blast.ncbi.nlm.nih.gov) [50]. The target sequence had at least two bases different from similar off-target sequences within the PAM or PAM-proximal region. The two gRNAs under the U3 and U6a promoters were introduced into the pYLCRISPR/Cas9 vector to generate the knockout construct pC-22-KT12. All the constructs were confirmed by sequencing. The primers used for polymerase chain reactions (PCRs) during pC-22-KT12 construction are presented in Appendix A.

The correct pC-22-KT12 vector was introduced into *Agrobacterium tumefaciens* EHA105 strains. Bacteria containing the pC-22-KT12 construct were transformed into the 733-3B line as previously described [51] or by Wuhan Biorun Biological Technology Co., Ltd., to generate T0 plants for salt-tolerant mutant screening. After four weeks of rooting, regenerated plants were transferred to plastic buckets and grown in a greenhouse at 28 °C for 14 h day and 26 °C for a 10 h night photoperiod.

### 4.3. Mutant Identification and Characterization in Edited Plants

T0 plants were collected from plates containing 50 mg/L G418 or 50 mg/L hygromycin. The T0 plants were then transferred to grow in a greenhouse, and the T0 seeds were harvested for further analysis. To confirm the targeted mutagenesis in the plants, genomic DNA (gDNA) was extracted using the cetyltrimethylammonium bromide (CTAB) method from a minimum of five leaves from different tillers during the mature period of growth [52]. The gDNA was used as the temple for target-PCR. PCR amplification with KOD FX (TOYOBO, Osaka, Japan) was performed using specific primers RR-CJ3F/RR-CJ3R (Appendix A) containing the CRISPR/Cas9 target sites. Following PCR, the products were purified and subjected to sequencing. Mutations were identified by comparing the T0 plant sequences to those of the wild-type. Plants were considered homozygous when mutations were associated with a normal sequencing chromatogram. Alternatively, mutations that exhibited superimposed sequencing chromatograms were considered heterozygous or biallelic. Samples with superimposed sequence chromatograms were cloned into the pEASY-Blut vector (TransGen Biotech, Beijing, China), and ten positive clones were sequenced to determine the mutation genotype. DNAMAN5.0 and MAGE4.0 were used for sequence alignment analysis.

### 4.4. Transgene-Free Detection

The identification of the absence of the transfer DNA (T-DNA) containing *Cas9* and *HPT* in the T1 generation was determined based on *Cas9-* and *HPT*-specific PCRs. The primer pairs Cas9-F/Cas9-R and HPT-F/HPT-R were used to amplify the *Cas9* and *HPT* genes, respectively (Appendix A). Plants were considered transgene-free when amplicons lacked both of these genes, with the pYLCRISPR/Cas9 plasmid and wild-type plants serving as positive and negative controls, respectively.

### 4.5. Measurement of Salinity Tolerance

A salt stress test was performed using the Takagi et al. [5] method to evaluate the salinity tolerance of *osrr22* lines at the seedling stage. Briefly, three-week-old *osrr22* and wild-type plants were selected and treated with Yoshida nutrient solution and 0.8% NaCl nutrient solution, respectively. The pH value of the nutrient solution was adjusted to 5.6–5.8, and the solution was renewed every 3 days. The plants were grown in a greenhouse at 28 °C for 12 h light and 22 °C for 12 h dark photoperiod. After 10 days of treatment with the 0.8% NaCl nutrient solution and 7 days of recovering with the nutrient solution, the different genotypes were photographed and compared. The salinity tolerance was determined by measuring the fresh weight, seedling length, root length, dry weight, and survival ratios of five plants per line. The measurements for each line were repeated in triplicate.

Whole growth stage trials were carried out in a salt-treated pool during the 2022 growing season in Sanya (a coastal city in southern China) to assess the salt tolerance of the promising third-generation hybrid rice line. Four-week-old seedlings of the new hybrid rice line and the control were transplanted. These were periodically irrigated with a mix of seawater (an approximately 1.0–1.5% NaCl concentration) and fresh water. The yield of these two hybrids was further analyzed. The soil salinity level in the seawater-treated plot was monitored by measuring the mixed soil’s electrical conductivity (EC) value, which was selected from five plots of the salt-treated pool using a Soli-EC tester (ZD-EC).

### 4.6. Amino Acid Sequence Analysis

The amino acid sequence of OsRR22 was deduced and aligned with the *OsRR22* mutants and wild-type using MAGE4.0 and DNAMAN5.0 software.

### 4.7. Agronomic Trait Characterization under Natural Growth Conditions

In order to assess the agronomic traits under normal conditions, both *osrr22* and wild-type plants were grown under standard conditions in a screenhouse in Changsha or Sanya. Four-week-old seedlings were transplanted, and management was conducted according to local conventional methods. Five plants in the middle rows of each line were sampled for the following traits: Plant height, the number of tillers per plant, the number of grains per panicle, spikelet fertility, and 1000-seed weight. These traits were all measured in five biological replicates per line.

### 4.8. Statistical Analysis

Three independent samples were taken from *osrr22* lines and wild-type to measure fresh weight, seedling length, root length, dry weight, and survival ratios. Five plants from each T4 population and wild-type were used for agronomic trait measurement. SPSS20.0 (IBM, Armonk, NY, USA) was used for the variance least significant difference test.

## Figures and Tables

**Figure 1 ijms-24-08025-f001:**
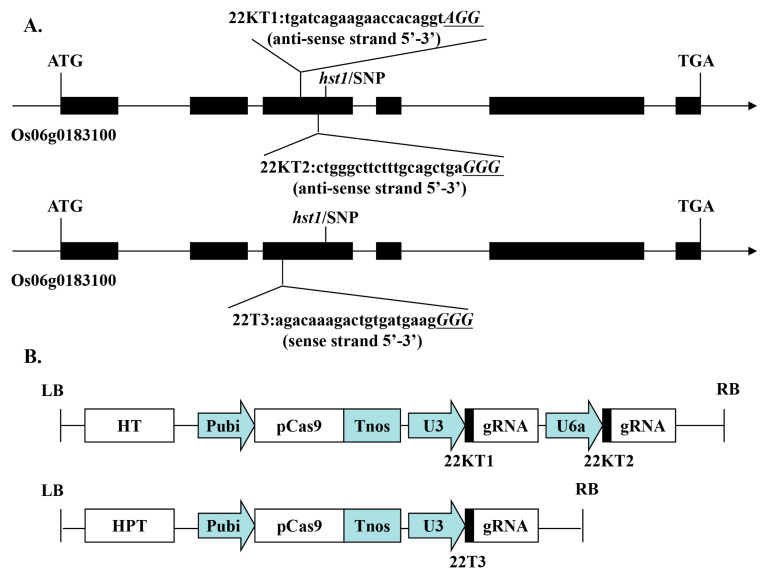
Target sites of the CRISPR/Cas9-OsRR22-KT12, 22T3. (**A**) Schematic of the *OsRR22* gene structure and target site. Exons and introns are indicated with black rectangles and black lines, respectively. Both the translation initiation codon (ATG) and the termination codon (TGA) are shown. The target sit nucleotides are shown in lowercase, and the protospacer adjacent motif (PAM) site is in uppercase, italic and underline; (**B**) schematic presentation of the T-DNA structure in the CRISPR/Cas9-mediated genome editing construct. The expression of Cas9 is driven by the Maize ubiquitin promoter (Pubi); the expression of the gRNA scaffold is driven by the rice U3 or U6a small nuclear RNA promoter (OsU3 or OsU6a); the expression of hygromycin (*HPT*) is driven by two CaMV35S promoters (2 × 35S); Tnos, gene terminator; LB and RB, left border and right border, respectively.

**Figure 2 ijms-24-08025-f002:**
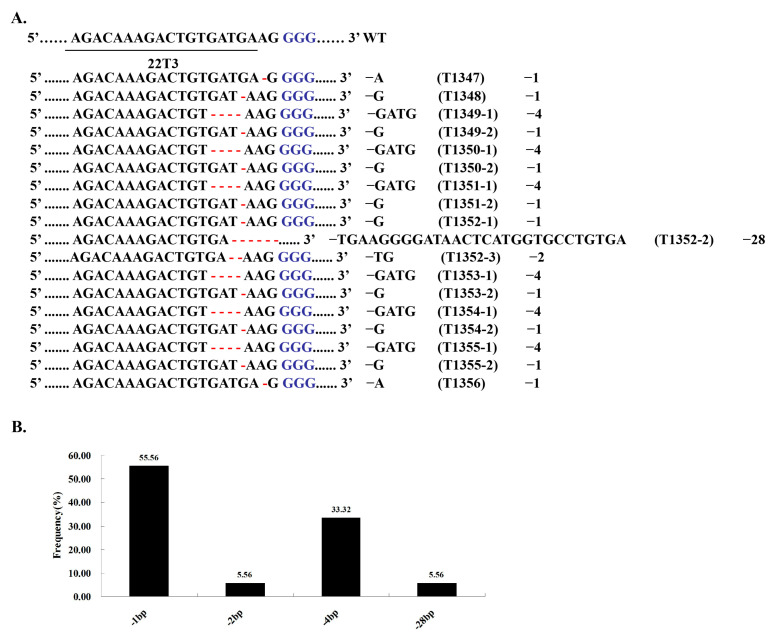
CRISPR/Cas9-induced *OsRR22* gene modification in rice. (**A**) Nucleotide sequences at the target site in the 10 T0 mutant rice plants. The recovered mutated alleles are shown below the wild-type sequence. The target site nucleotides are indicated by black capital letters. The PAM site is highlighted in blue. Mutations deletions are shown by red hyphens. The numbers on the right indicate the type of mutation and the number of nucleotides involved. “–” indicates the deletion of the indicated number of nucleotides; “WT” indicates wild-type; (**B**) in all types of induced mutations, single-nucleotide insertions were most frequently detected.

**Figure 3 ijms-24-08025-f003:**
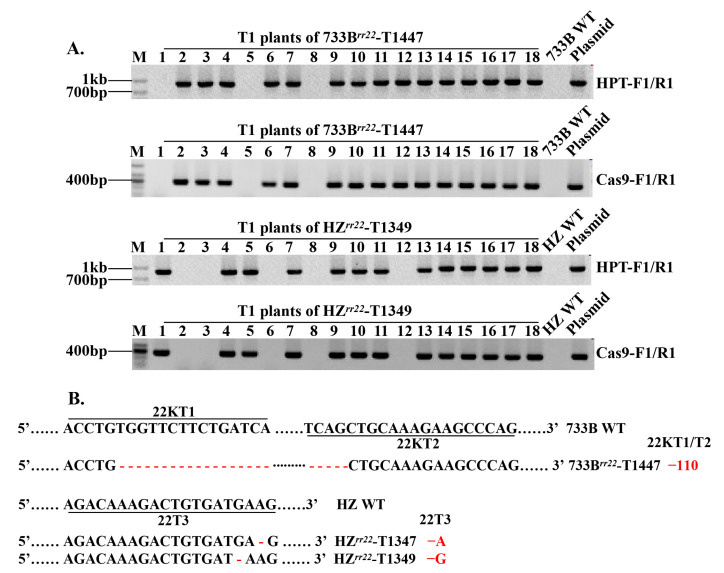
Mutation types and PCR-based identification of transgene-free in the T1 generation. (**A**) *HPT* and *Cas9* gene-specific PCR amplification of 733B*^rr22^*-T1447 and HZ*^rr22^*-T1349; (**B**) the mutation types of 733B*^rr22^*-T1447, HZ*^rr22^*-T1347 and HZ*^rr22^*-T1349. Mutations deletions are shown by red hyphens. The numbers on the right indicate the type of mutation and the number of nucleotides involved. “−” indicates the deletion of the indicated number of nucleotides; “WT” indicates wild-type.

**Figure 4 ijms-24-08025-f004:**
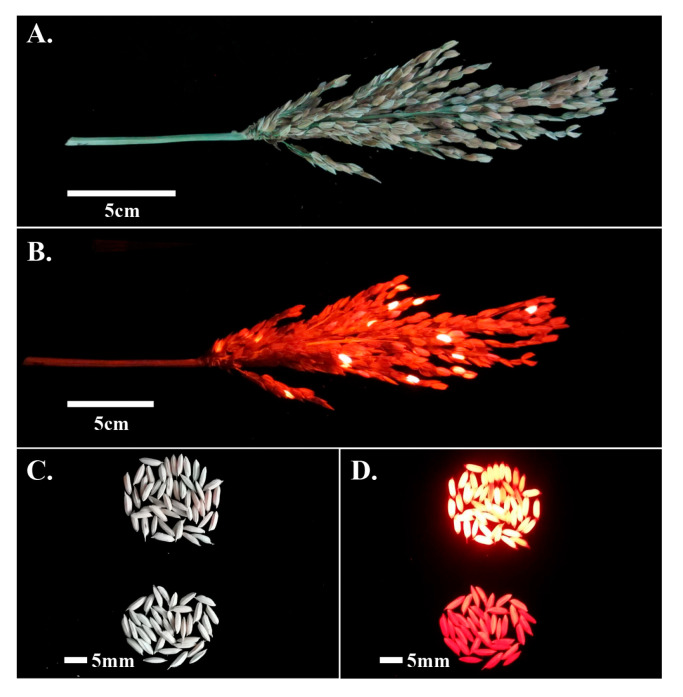
The seed fluorescence of the *OsRR22* mutagenesis maintainer line 733B*^rr22^*-T1447-1; (**A**,**B**) Panicle phenotype of the *osrr22* maintainer line 733B*^rr22^*-T1447-1 under bright field (BF) and red fluorescent field (RFP); (**C**,**D**) grain phenotype of the *osrr22* maintainer line 733B*^rr22^*-T1447-1 under bright field (BF) and red fluorescent field (RFP).

**Figure 5 ijms-24-08025-f005:**
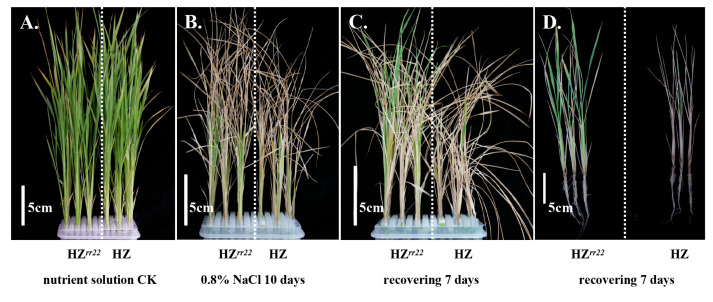
Identification of salinity tolerance in *osrr22* mutant lines. HZ*^rr22^*: HZ*^rr22^*-T1349-3, HZ: HZ-WT. (**A**) Phenotypes of 4.5-week-old HZ*^rr22^* and HZ-WT plants grown with underground Yoshida nutrient solution; (**B**) phenotypes of 4.5-week-old HZ*^rr22^* and HZ-WT plants grown with underground 0.8% NaCl nutrient solution. 3-week-old plants were treated with 0.8% NaCl nutrient solution, then phenotypic evaluation was performed 10 days after treatment; (**C**,**D**) phenotypes of 5.5-week-old HZ*^rr22^* and HZ-WT plants grown with underground Yoshida nutrient solution. After 0.8% NaCl nutrient solution treatment, plants recovered via fresh nutrient solution for 7 days.

**Figure 6 ijms-24-08025-f006:**
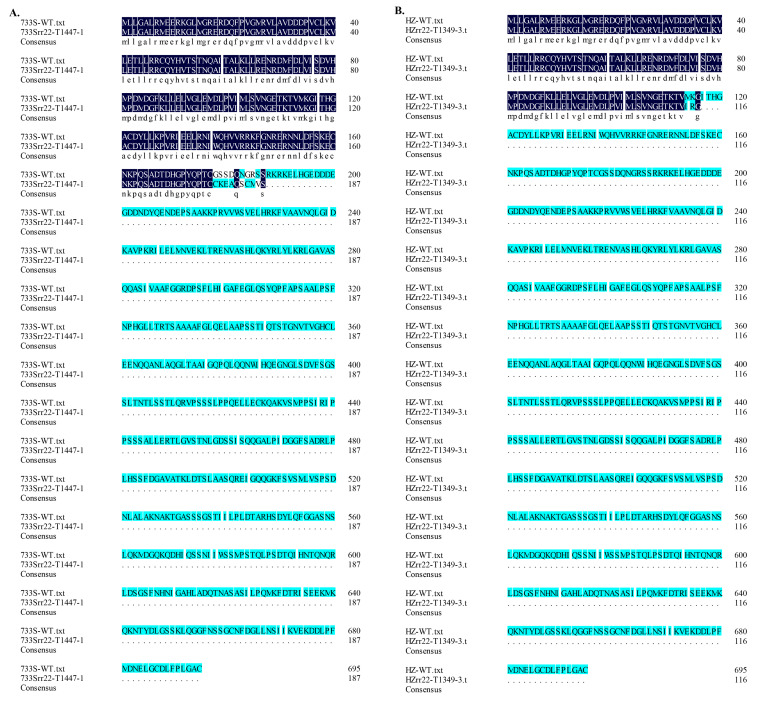
The amino acid sequence of *rr22* transgene-free homozygous mutant lines. Deduced OsRR22 amino acid sequence alignment for the two transgene-free homozygous mutant lines (733S*^rr22^*-T1447-1, HZ*^rr22^*-T1349-3) and for WT lines (733S, HZ). Each of the mutant alleles code for truncated and disrupted OsRR22 proteins. Black highlights are the same part of deduced OsRR22 amino acid sequence between homozygous mutant lines and WT lines. Blue highlights are the different part of deduced OsRR22 amino acid sequence between homozygous mutant lines and WT lines. (**A**) Deduced OsRR22 amino acid sequence alignment for the 733S*^rr22^*-T1447-1 and 733S; (**B**) Deduced OsRR22 amino acid sequence alignment for the HZ*^rr22^*-T1349-3 and HZ.

**Figure 7 ijms-24-08025-f007:**
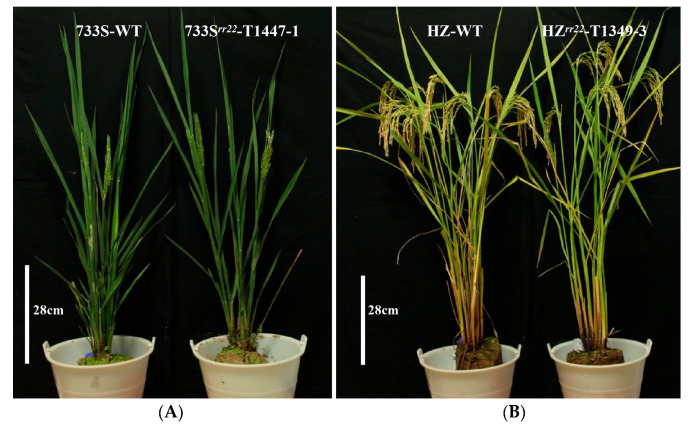
Agronomic traits of 733S*^rr22^*-T1447-1, HZ*^rr22^*-T1349-3 T4 progeny, and WT plants. (**A**,**B**) T4 progeny of 733S*^rr22^*-T1447-1, HZ*^rr22^*-T1349-3, and WT (733S-WT, HZ-WT) are displayed.

**Figure 8 ijms-24-08025-f008:**
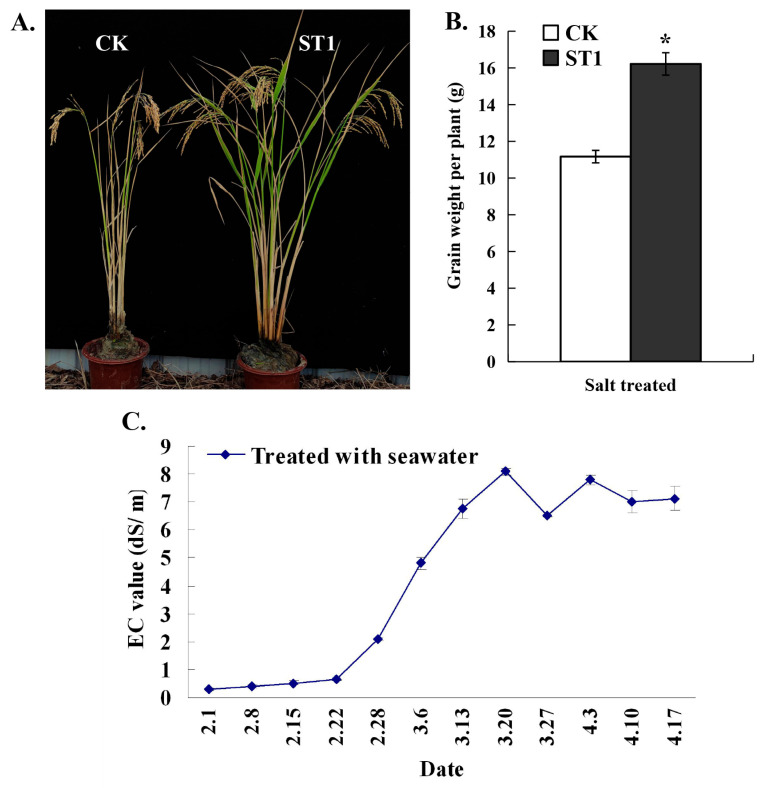
Evaluation of the growth performance and yield of CK and ST1 plants under seawater-treated conditions. (**A**) The phenotypes of salt-treated CK and ST1 plants; (**B**) Grain yield of CK and ST1 plants. The histogram marked with the “*” is significantly different (*p* < 0.05, Student’s *t*-test); (**C**) Electrical conductivity (EC) values measured over the two-month growing period in the plot used for evaluation of CK and ST1 plants used in (**A**).

**Table 1 ijms-24-08025-t001:** Performance of salinity tolerance of *osrr22* mutant lines at seedling stage.

Line	Seedling Length (cm)	Root Length (cm)	Fresh Weight (g)	Dry Weight (mg)	Survival Ratios (%)
733S-WT	16.67 ± 0.40	9.67 ± 1.40	0.25 ± 0.05	68.67 ± 10.02	33.81 ± 0.84
733S*^rr22^*-T1447-1-T3	22.17 ± 0.76 *	11.63 ± 2.41	0.92 ± 0.18 *	183.00 ± 9.54 *	72.05 ± 3.34 *
HZ-WT	21.93 ± 0.95	5.70 ± 1.04	0.15 ± 0.03	57.00 ± 7.55	29.89 ± 3.00
HZ*^rr22^*-T1347-6-T3	24.63 ± 1.33 *	9.47 ± 0.90 *	0.46 ± 0.06 *	86.00 ± 8.00	47.69 ± 2.28 *
HZ*^rr22^*-T1349-3-T3	29.53 ± 0.92 *	10.40 ± 1.22 *	0.73 ± 0.07 *	185.00 ± 10.82 *	61.48 ± 2.79 *

Values are mean ± SD of 3 biological replicates. Means were compared by Student’s *t*-test. * significantly different (*p* < 0.05, Student’s *t*-test).

**Table 2 ijms-24-08025-t002:** Agronomic traits of 733S*^rr22^*-T1447-1, HZ*^rr22^*-T1349-3 T4 progeny, and WT plants.

Lines	Plant Height (cm)	The Number of Tillers per Plant	The Number of Grains per Panicle	Spikelet Fertility (%)	1000-Seed Weight (g)
733S-WT	92.80 ± 3.24 ^a^	4.20 ± 0.45 ^a^	557.00 ± 52.63 ^a^	-	-
733S*^rr22^*-T1447-1	92.40 ± 1.31 ^a^	4.40 ± 0.55 ^a^	542.00 ± 34.58 ^a^	-	-
HZ-WT	116.34 ± 2.34 ^b^	11.60 ± 1.51 ^b^	241.60 ± 17.24 ^b^	78.47 ± 4.18 ^b^	17.37 ± 0.35 ^b^
HZ*^rr22^*-T1349-3	117.15 ± 1.26 ^b^	11.20 ± 1.34 ^b^	217.40 ± 10.97 ^b^	76.25 ± 3.73 ^b^	17.13 ± 0.16 ^b^

The results are shown for five plants of each mutant line and are represented as the mean ± SD. Means were compared by Student’s *t*-test, and the values marked with the same letter (a/b) are non-significantly different (*p* > 0.05, Student’s *t*-test).

## Data Availability

All data generated or analyzed during this study are included in this published article and its Appendix A.

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
