# Peer review of "Novel Salinity-Tolerant Third-Generation Hybrid Rice Developed via CRISPR/Cas9-Mediated Gene Editing"

_ijms, 2023, doi:10.3390/ijms24098025_

Round 1

Reviewer 1 Report

In this manuscript author did “Novel salinity-tolerant third-generation hybrid rice developed via CRISPR/Cas9-mediated gene editing”. In this study aimed to improve rice salinity tolerance by combining targeted CRISPR/Cas9-mediated editing of the OsRR22 gene with heterosis utilization. The novel alleles of the genic male-sterility (GMS) and elite restorer line (733Srr22-T1447-1 and HZrr22-T1349-3) produced 110 and 1 bp deletions at the third exon of OsRR22 and conferred a high level of salinity-tolerance. Homozygous transgene-free progeny were identified via segregation in the T2 generation with osrr22 showing similar agronomic performance to wild-type (733S and HZ). Furthermore, these two osrr22 lines were used to develop a new promising third-generation hybrid rice line with novel salinity tolerance. Overall, the results demonstrate that combining CRISPR/Cas9 targeted gene editing with the “third-generation hybrid rice system” approach allows for the efficient development of novel hybrid rice cultivars that exhibit a high level of salinity tolerance, thereby ensuring improved cultivar stability and enhanced rice productivity.

The manuscript is written very well and covers all the spect. Therefore, the manuscript can be accepted in its current format.

Reviewer 2 Report

I am unable to review the current manuscript in its current form due to the absence of all figures and tables. I would like to ask authors to resubmit the manuscript. 

Reviewer 3 Report

Salinity is one of the major abiotic stressors negatively affecting crop growth and development. in this study, the pC-22-KT12 and pC-22-T3 Cas9 vectors were used to edit OsRR22. The salinity tolerance evaluations at the seedling stage demonstrated that the salinity tolerance was markedly enhanced in two novel OsRR22 alleles (733Srr22-T1447-1 and HZrr22-T1349-3), suggesting that these alleles are valuable resources for developing elite rice varieties that can be cultivated under salinity conditions. The positive results suggest that integrating both the CRISPR/Cas9280 approach and heterosis utilization may potentially represent a powerful, highly efficient, and green approach to the genetic improvement of rice and other hybrid crop breeding. As a result, this study offers a new strategy and novel OsRR22 resources conferring high salinity tolerance and desirable agronomic traits for salt tolerance breeding via combined CRISPR/Cas9 technology and a third-generation hybrid rice system. This paper is very innovative.

The relationship between Fig. 5 – A-D should be specified in the text – chapter 2.3. Novel OsRR22 alleles conferring salt-tolerance.

Round 2

Reviewer 1 Report

it can be accepted in its current form.

Author Response

Thank you for guiding us revise and improve the quality of our paper.

Reviewer 2 Report

Please find below my main concerns/suggestions:

1: Authors must include the results of T3 generation (minimum criteria).

2: There are many linguistic mistakes in the edited version. 

3: Add yield per hectare (Figure 8B) by utilizing grain weight per plant

4: Discussion is very poor. There is no logical reasoning. Please revise the discussion with recent literature in a scientific and logical way to support your results. Avoid biological inferences.

5: Most importantly, significance of the study is missing in introduction. Add more information from recent literature about the effects of salinity stress on rice. Why salinity is an important aspect to be studied in rice
